# Preparation of CaCO_3_/Al(OH)_3_ Composites via Heterogeneous Nucleation

**DOI:** 10.3390/ma16020498

**Published:** 2023-01-04

**Authors:** Yan Xu, Weijun Bao, Hao Ding, Jingkui Qu

**Affiliations:** 1Engineering Research Center of Ministry of Education for Geological Carbon Storage and Low Carbon Utilization of Resources, Beijing Key Laboratory of Materials Utilization of Nonmetallic Minerals and Solid Wastes, National Laboratory of Mineral Materials, School of Materials Science and Technology, China University of Geosciences Beijing, Beijing 100083, China; 2Institute of Process Engineering, Chinese Academy of Sciences, Beijing 100190, China

**Keywords:** calcium carbonate, aluminum hydroxide, CaCO_3_/Al(OH)_3_ composite, heterogeneous nucleation, preparation

## Abstract

As one of the most widely used inorganic fine powder fillers, calcium carbonate is cheap. However, considering its poor light transmittance, it is not suitable to be added to resin matrix composites that require high light transmittance. Aluminum hydroxide has good light transmission and flame retardancy, but it is more expensive than calcium carbonate. CaCO_3_/Al(OH)_3_ composites with a core-shell structure that showed a trend toward the performance of aluminum hydroxide not only improved the surface properties of CaCO_3_, but also increased the added value of CaCO_3_. In the present paper, CaCO_3_/Al(OH)_3_ composites were successfully prepared in sodium aluminate solution via heterogeneous nucleation. Four types of calcium sources, including calcite-type precipitated calcium carbonate, vaterite-type precipitated calcium carbonate, ground calcium carbonate with two different particle sizes as the precursors and supersaturated sodium aluminate solution as the substrate, have been deeply investigated in terms of their influence on the preparation of CaCO_3_/Al(OH)_3_ composites. Results showed that the calcium carbonate precursor greatly affected the formation of CaCO_3_/Al(OH)_3_ composites. Both the precipitated calcium carbonate and the small particle ground calcium carbonate are likely to undergo anti-causticization and a complexation reaction with it to generate 3CaO·Al_2_O_3_·6H_2_O and 3CaO·Al_2_O_3_·CaCO_3_·11H_2_O, which go against the coating of calcium carbonate with aluminum hydroxide. Within the experimental range, the use of ground calcium carbonate with a particle size of 400–500 mesh is more suitable as a precursor for the preparation of core-shell CaCO_3_/Al(OH)_3_ composites.

## 1. Introduction

Calcium carbonate (CaCO_3_) is one of the world’s most abundant inorganic materials [1], and it is cheap and has wide industrial applications [2,3,4], especially in the field of inorganic fine powder fillers [5,6,7]. A large amount of high-quality white marble mine resources are present in the Hezhou Region of Guangxi Province in China, and these resources are the largest in the South China Region with prospective reserves of as much as 2.6 billion cubic meters [8]. The technical indexes of the white marble resources can meet the special grade level in China, in which the calcium carbonate content is above 96% and the whiteness is above 95. However, for the resource utilization of marble, the proportion of value-added products is low, and it is usually used to prepare common ground calcium carbonate or the fillers of artificial marble [9]. The use pattern decreases the value of the resource in bringing out serious waste, and it cannot achieve sustainable development. A new way needs to be explored for enhancing the additional value of calcium carbonate.

Generally, aluminum hydroxide (Al(OH)_3_) is a chemical alumina variety and a refined mineral filler with a relatively stable performance [10]. Aluminum hydroxide, which has similar application fields to those of calcium carbonate with high added value, is much more expensive than ordinary calcium carbonate, and it has good light transmission and flame retardancy [11,12]. It is mainly used as a flame-retardant additive to plastic rubber in wire, cable, insulators, copper clad laminate, prepared man-made gems and acrylic boards [13,14,15]. The market consumption of aluminum hydroxide has reached 1 million tons.

In CaCO_3_/Al(OH)_3_ composites, the surface of calcium carbonate is coated with aluminum hydroxide, which has properties that are similar to those of aluminum hydroxide. They can grant calcium carbonate the properties of aluminum hydroxide and increase its value. Nowadays, the preparation of CaCO_3_/Al(OH)_3_ composites has been drawing increased attention. OTA et al. [16] reported a type of CaCO_3_/Al(OH)_3_ composites, which involve aragonite CaCO_3_ fibers coated with fine particles of Al(OH)_3_. The synthetic method for the preparation of this composite includes the use of NaAlO_2_ as the aluminum source and Ca(OH)_2_ as the calcium source with a carbonation method. Al(OH)_3_ can be coated on a CaCO_3_ whisker by setting the pH value of the synthetic system above 10.5. Ding et al. [17] reported a method for the synthesis of CaCO_3_/Al(OH)_3_ composites involving calcite spindly calcium carbonate coated with AlOOH, in which Al_2_(SO_4_)_3_ was used as the aluminum source and Ca(OH)_2_ was used as the calcium source via a bubbling carbonization method. This process includes two steps. First, calcium carbonate slurry with a pH of 8–9 was obtained by carbonization of calcium hydroxide slurry with blowing CO_2_ gas. Thereafter, CaCO_3_/AlOOH composites were obtained by adding the Al_2_(SO_4_)_3_ aqueous solution into the slurry while adjusting the pH of the suspension to 8–10.

The above-reported CaCO_3_/Al(OH)_3_ composites were prepared from Ca(OH)_2_ as a calcium source by carbonation, and the pH of the synthetic system must be precisely controlled. If the ordinary calcium carbonate can be directly used for the preparation of CaCO_3_/Al(OH)_3_ composites, the technological process can be simplified and energy consumption can be reduced. In this paper, CaCO_3_/Al(OH)_3_ composites were successfully prepared with CaCO_3_ as a calcium source in a sodium aluminate solution via heterogeneous nucleation. Four types of calcium sources, including calcite-type precipitated calcium carbonate, vaterite-type precipitated calcium carbonate, and ground calcium carbonate with two different particle sizes as the precursors and supersaturated sodium aluminate solution as the substrate, have been deeply investigated in terms of their influence on the preparation of CaCO_3_/Al(OH)_3_ composites. Accordingly, a suitable calcium source precursor of the preparation of CaCO_3_/Al(OH)_3_ composites was selected.

## 2. Materials and Methods

### 2.1. Material and Reagents

Sodium hydroxide (NaOH), aluminum hydroxide (Al(OH)_3_), calcium chloride (CaCl_2_), calcium acetate ((CH_3_COO)_2_Ca), sodium carbonate (Na_2_CO_3_) and sucrose (C_12_H_22_O_11_) were obtained from Sinopharm Chemical Reagent Co., Ltd., China. All the reagents used were of analytical grade. The ordinary ground calcium carbonate powder was obtained from a factory in Hezhou city Guangxi Province in China. Deionized water was applied throughout this study.

### 2.2. Preparation of CaCO_3_/Al(OH)_3_ Composites

The preparation of CaCO_3_/Al(OH)_3_ composites includes the preparation of different calcium source precursors, the preparation of a supersaturated sodium aluminate solution and the synthesis of CaCO_3_/Al(OH)_3_ composites by heterogeneous nucleation.

#### 2.2.1. Preparation of Different Calcium Source Precursors

Vaterite-type precipitated calcium carbonate was prepared by double decomposition according to the following steps. First, 148.6 g Na_2_CO_3_ was dissolved in 14 L of deionized water (30 °C, 300 rpm) in a 20 L stirred double glazed reaction kettle and 221.5 g (CH_3_COO)_2_Ca was dissolved in 0.9 L of deionized water in a 2 L glass beaker. Then, the prepared (CH_3_COO)_2_Ca aqueous solution was added into a stirred double glazed reaction kettle filled with Na_2_CO_3_ aqueous solution with a peristaltic pump at a constant speed of 90 mL/min. The obtained CaCO_3_ suspension was filtered using a vacuum pump. The filtered cake was washed thrice by using deionized water. The washed solid residue was dried at 105 °C for 4 h in the oven.

Calcite-type precipitated calcium carbonate was prepared by way of a double decomposition reaction according to the following steps. First, 190.8 g Na_2_CO_3_ was dissolved in an aqueous solution (deionized water 18 L) in a 20 L stirred double glazed reaction kettle and 199.8 g CaCl_2_ was dissolved in 0.9 L of deionized water in a 2 L glass beaker. Then, the prepared CaCl_2_ aqueous solution was added into the stirred double glazed reaction kettle filled with an Na_2_CO_3_ aqueous solution by using the peristaltic pump at a constant speed of 90 mL/min. The obtained CaCO_3_ slurry was aged for 12 h and filtered using a vacuum pump. The filtered cake was washed until Cl^-^ was eliminated in the filtrate by using deionized water. The washed filtered cake was dried at 105 °C for 4 h in the oven.

Two different particle sizes of ground calcium carbonate were obtained using the wet-sieving method. Approximately 2 kg of ground calcium carbonate powder was mixed well with 15 L of water in a 20 L plastic bucket. Then, the slurry was screened using a stain-less steel sieve with different screen-apertures, such as 400 and 500 mesh. The particle size of 400–500 mesh and less than 500 mesh calcium carbonates, which were dried at 105 °C for 4 h, were selected as ground calcium carbonate source precursors.

#### 2.2.2. Preparation of Supersaturated Sodium Aluminate Solution

Supersaturated sodium aluminate solution was made by dissolving Al(OH)_3_ into a hot NaOH solution at boiling point. First, 271 g NaOH was dissolved in 2 L deionized water in a 3 L stainless steel tank and was heated to boiling point. Then, 330.4 g of Al(OH)_3_ was dissolved into a hot NaOH solution and was heated continuously until the mixture was visibly pellucid. Then, the hot solution was vacuum filtered twice through double-layer filter paper and stored in a caustic-resistant polyethylene vessel. The initial sodium oxide concentration and aluminum oxide concentration of prepared sodium aluminate solution was about 105 g/L and 108 g/L, respectively. The solution was used as soon as it was ready.

#### 2.2.3. Preparation of CaCO_3_/Al(OH)_3_ Composites

CaCO_3_/Al(OH)_3_ composites were prepared by heterogeneous nucleation. First, 300 mL of the prepared supersaturated sodium aluminate solution was added into a 500 mL stirred three-necked flask and heated in a constant temperature water bath at 80 °C. Approximately 13.8 g calcium carbonate was then added to the solution under stirring. The stirring speed was 400 rpm. Subsequently, the experimental temperature was lowered to 70 °C over 4 h and was kept constant for 32 h. Finally, the slurry was filtrated, the precipitate was washed with deionized water to neutralize it, and it was dried at 105 °C for 4 h. The previous steps were repeated, and a series of the CaCO_3_/Al(OH)_3_ composite samples was obtained by changing the calcium source precursor.

### 2.3. Material Characterization

The crystal structure of samples was identified by X-ray diffraction (XRD, Smartlab (9), Rigaku, Tokyo, Japan) with Cu Kα radiation (λ = 1.5406 Å) in the range of 10°–70° with a step size of 0.01°/step under 200 mA and 40 kV. The morphology was determined via field emission scanning electron microscopy (SEM, JSM-7610F, JEOL, Tokyo, Japan). The conductivity of the samples was improved by spray-gold treatment. Cross-sectional image and the elemental distribution of the sample were obtained by mixing the sample well with resin (the mass ratio of samples/resin 1:1) in a 25 mm-wide cylindrical silicon mold. After curing at room temperature, a cross-section of the sample was exposed by polishing the metallographic specimen with diamond slurry (water-based) in a UniPOL 202D polishing machine. The sample was imaged and characterized by field emission scanning electron microscopy (SEM, JSM-7610F, JEOL, Tokyo, Japan) equipped with an energy-dispersive spectrometer (EDS, Ultim Max, Oxford Instruments, Oxford, UK).

## 3. Results and Discussion

### 3.1. Crystal Phase Analysis

Figure 1 shows the XRD patterns of different calcium sources and their composite samples. Figure 1a,b show that the XRD pattern of 400–500 mesh ground calcium carbonate shows strong characteristic peaks of the calcite crystalline phase ((JCPDS #99-0022, 2θ = 23.06°, 29.40°, 35.97°, 39.41°, 43.16°, 47.50°, 48.51°, and 57.40°). Its composite sample shows additional diffraction peaks at 17.28° and 18.50°, which can be ascribed to the (211) crystalline planes of the 3CaO·Al_2_O_3_·6H_2_O (C_3_AH_6_) phase [18] (JCPDS #77-0240) and (002) crystalline planes of the Al(OH)_3_ (AH_3_) phase (JCPDS #85-1049). As shown in Figure 1c,d, the ground calcium carbonate with a particle size less than 500 mesh also shows diffraction peaks at 23.06°, 29.40°, 35.97°, 39.41°, 43.16°, 47.50°, 48.51°, and 57.40°, which can be attributed to the crystalline planes of the calcite phase. The ground calcium carbonate composite sample with a particle size of less than 500 mesh showed new strong diffraction peaks at 17.28° and 18.50°, which can be ascribed to the C_3_AH_6_ and Al(OH)_3_ (AH_3_) phases, respectively. In addition, only a weak peak was observed at 11.71°, which belongs to the (011) crystalline planes of the 3CaO·Al_2_O_3_·CaCO_3_·11H_2_O (C_4_AcH_11_) phase [19] (JCPDS #87-0493).

Figure 1e,f display the XRD patterns of the precipitated calcium carbonate that was prepared from the CaCl_2_ solution and its composite with Al(OH)_3_. The diffraction peaks at 23.06°, 29.40°, 35.97°, 39.41°, 43.16°, 47.50°, 48.51°, and 57.40° can be attributed to the crystalline planes of the calcite phase, while those at 11.71°, 17.28°, and 18.50° can be attributed to the crystalline planes of C_4_AcH_11_, Al(OH)_3_, and C_3_AH_6_, respectively. In comparison with the ground calcium carbonate composite with a particle size of less than 500 mesh, the diffraction peaks at 11.71° were obviously stronger, indicating that the composite from calcite-type precipitated calcium carbonate contains more C_4_AcH_11_ phase. Figure 1g shows the XRD pattern of the precipitated calcium carbonate that was prepared from the (CH_3_COO)_2_Ca solution. It shows diffraction peaks at 21.00°, 24.90°, 27.05°, 32.78°, 43.85°, 50.08°, and 55.81°, which can be ascribed to the (004), (110), (112), (114), (300), (118), and (224) crystalline planes of the vaterite phase [20,21] (JCPDS #72-0506). As shown in Figure 1h, the vaterite-type precipitated calcium carbonate composite also contained extra diffraction peaks at 11.71°, 17.28°, and 18.50°, which can be ascribed to the crystalline planes of C_4_AcH_11_, C_3_AH_6_ and AH_3_ phases. However, the intrinsic peaks belonging to the vaterite phase at 21.00°, 24.90°, 43.85°, 50.08°, and 55.81° vanished and were greatly reduced at 27.05° and 32.78°, indicating that most of the vaterite-type precipitated calcium carbonate was converted to another phase.

Based on the above results, only two or three new characteristic peaks appeared in the four types of CaCO_3_/Al(OH)_3_ composites, including calcite, vaterite, C_4_AcH_11_, C_3_AH_6_ and AH_3_ crystal phases. The weight percentage of each phase in four types of CaCO_3_/Al(OH)_3_ composites was determined using MDI Jade 6 based on the matrix-flushing theory [22], the adiabatic principle [23] and the RIR-value in the JCPDS card according to the XRD patterns. The mass fraction of phase *X* was calculated as follows:
(1)$$\omega_{X}=\frac{I_{X_{i}}}{K_{A}^{X}\sum_{i=A}^{N}\frac{I_{i}}{K_{A}^{i}}},$$
where $\omega_{X}$ is the mass fraction of the *X* phase, $I_{X_{i}}$ is the integral intensity of the strongest peak in the *X* phase, $K_{A}^{X}$ is the *K*-value of the *X* phase, and *N* is the number of sample phases.

Taking the CaCO_3_ phase as the standard phase, according to formula (1), the mass fraction of each phase can be calculated as follows:


(2)
$$\omega_{C_{4}AcH_{11}}=\frac{I_{C_{4}AcH_{11}}}{K_{CaCO_{3}}^{C_{4}AcH_{11}}\left(\frac{I_{C_{4}AcH_{11}}}{K_{CaCO_{3}}^{C_{4}AcH_{11}}}+\frac{I_{C_{3}AH_{6}}}{K_{CaCO_{3}}^{C_{3}AH_{6}}}+\frac{I_{AH_{3}}}{K_{CaCO_{3}}^{AH_{3}}}+\frac{I_{CaCO_{3}}}{K_{CaCO_{3}}^{CaCO_{3}}}\right)},$$



(3)
$$\omega_{C_{3}AH_{6}}=\frac{I_{C_{3}AH_{6}}}{K_{CaCO_{3}}^{C_{3}AH_{6}}\left(\frac{I_{C_{4}AcH_{11}}}{K_{CaCO_{3}}^{C_{4}AcH_{11}}}+\frac{I_{C_{3}AH_{6}}}{K_{CaCO_{3}}^{C_{3}AH_{6}}}+\frac{I_{AH_{3}}}{K_{CaCO_{3}}^{AH_{3}}}+\frac{I_{CaCO_{3}}}{K_{CaCO_{3}}^{CaCO_{3}}}\right)},$$



(4)
$$\omega_{AH_{3}}=\frac{I_{AH_{3}}}{K_{CaCO_{3}}^{AH_{3}}\left(\frac{I_{C_{4}AcH_{11}}}{K_{CaCO_{3}}^{C_{4}AcH_{11}}}+\frac{I_{C_{3}AH_{6}}}{K_{CaCO_{3}}^{C_{3}AH_{6}}}+\frac{I_{AH_{3}}}{K_{CaCO_{3}}^{AH_{3}}}+\frac{I_{CaCO_{3}}}{K_{CaCO_{3}}^{CaCO_{3}}}\right)},$$



(5)
$$\omega_{CaCO_{3}}=\frac{I_{CaCO_{3}}}{K_{CaCO_{3}}^{CaCO_{3}}\left(\frac{I_{C_{4}AcH_{11}}}{K_{CaCO_{3}}^{C_{4}AcH_{11}}}+\frac{I_{C_{3}AH_{6}}}{K_{CaCO_{3}}^{C_{3}AH_{6}}}+\frac{I_{AH_{3}}}{K_{CaCO_{3}}^{AH_{3}}}+\frac{I_{CaCO_{3}}}{K_{CaCO_{3}}^{CaCO_{3}}}\right)}.$$


According to the XRD patterns as shown in Figure 1, the characteristic peaks at 11.7°, 17.3°, 18.5°, 27.0°, and 29.4° with a diffraction angle of 2θ represent the strongest diffraction peaks of C_4_AcH_11_, C_3_AH_6_, AH_3_, vaterite-CaCO_3_, and calcite-CaCO_3_, respectively. Therefore, the relative *K* value of those phases, including C_4_AcH_11_, C_3_AH_6_, AH_3_, vaterite-CaCO_3_, and calcite-CaCO_3_, could be calculated, and the results are shown in Table 1.

Figure 2 displays the content of each phase in four types of CaCO_3_/Al(OH)_3_ composites. The content of C_4_AcH_11_ in the composite obtained from precipitated calcium carbonate was much higher than that from the ground calcium carbonate composite, and the content of AH_3_ in the precipitated calcium carbonate composites was relatively higher than that of the ground calcium carbonate composites, while the content of CaCO_3_ was low. Moreover, four types of composites have C_3_AH_6_, while the contents of C_3_AH_6_ in the composite obtained from the ground calcium carbonate with a particle size of less than 500 mesh was relatively higher than that of the others. Therefore, the formed CaCO_3_/Al(OH)_3_ composites have a very different phase content. The CaCO_3_ molecule of vaterite-type calcium carbonate is more active than those of the three other types of calcium carbonates [24,25], and the complex reaction between C_3_AH_6_ and vaterite-type calcium carbonate easily occurs compared with others. Therefore, a large number of C_4_AcH_11_ exist in vaterite-type calcium carbonate composite. Calcite-type precipitated calcium carbonate has more active CaCO_3_ molecules in the solution than the <500 mesh ground calcium carbonate. Therefore, the content of C_4_AcH_11_ in calcite-type precipitated calcium carbonate composites is higher than that that in <500 mesh ground calcium carbonate composites. Additionally, with more C_3_AH_6_ converted to C_4_AcH_11_, the phase content of C_3_AH_6_ in calcite-type precipitated calcium carbonate composites is lower than that in <500 mesh ground calcium carbonate composites. Moreover, only a small amount of active CaCO_3_ molecule is present in solution, and the phase of C_4_AcH_11_ could hardly be detected for 400–500 mesh ground calcium carbonate composite.

### 3.2. Morphological Analysis

The SEM images of four calcium carbonate precursors, including vaterite-type precipitated calcium carbonate, calcite-type precipitated calcium carbonate, <500 mesh ground calcium carbonate and 400–500 mesh ground calcite carbonate are presented in Figure 3. Figure 3a shows that the vaterite-type precipitated calcium carbonate is composed of particles with aggregated and stacked spherical or ellipsoid structures. The single particle has a size of 1–5 μm, and the aggregate size is 20–30 μm. Figure 3b shows that the calcite-type precipitated calcium carbonate is composed of particles with a cubic crystalline structure. These particles are uniform and interleaved with each other; the average particle size is 10–20 μm. Figure 3c shows that the <500 mesh ground calcium carbonate is composed of polyhedron particles with different shapes and sizes. The particle size has a wide range from several microns to tens of microns. The bigger particles are stacked together in lamellar form, and many fine particles are adhered to their surfaces. In comparison with the <500 mesh ground calcium carbonate, the 400–500 mesh ground calcium carbonate is also composed of irregular polyhedrons, as shown in Figure 3d, while the particle size distribution is much narrower with a size of 25–30 μm. Therefore, the four calcium carbonate precursors have a regular morphology and similar particle size distribution but very different morphology.

The SEM images of vaterite-type precipitated calcium carbonate composite, calcite-type precipitated calcium carbonate composite, <500 mesh ground calcium carbonate composite and 400–500 mesh ground calcium carbonate composite are shown in Figure 4. Figure 4a shows that the particle morphology of vaterite-type precipitated calcium carbonate composite is irregular, and the surface consists of many sheets that are stacked with each other and a small amount of granular structure. In comparison with Figure 3a, the particle morphology changed remarkably, and the spherical and ellipsoid structure disappeared, indicating that the vaterite-type precipitated calcium carbonate has been converted into other species. Combined with XRD analysis, as shown in Figure 1h, the formed sheet may refer to AH_3_ or C_4_AcH_11_, and the granular structure may refer to C_3_AH_6_, as reported by Li [26], Litwinek [19] and Shang [27]. Figure 4b shows that the surface of calcite-type precipitated calcium carbonate composite is composed of lamellar, granular, and prismatic morphology, while the cubic crystalline structure morphology cannot be found. The lamellar, prismatic morphology can refer to AH_3_ or C_4_AcH_11_, and the granular morphology can refer to C_3_AH_6_. Figure 4c shows that the <500 mesh ground calcium carbonate composites contain both large and small particles, and the surface of the large particles has a granular and columnar morphology, which can also refer to C_3_AH_6_ or AH_3_, respectively. Figure 4d shows that the 400–500 mesh ground calcium carbonate composites only contain large particles, and the surface of the large particles has a prismatic morphology and conical shape structure, which is very similar to the morphology of AH_3_. The 400–500 mesh ground calcium carbonate particles are tightly wrapped by aluminum hydroxide.

Based on the above SEM images, the initial conclusion can be made that four types of calcium carbonate are coated with newly formed species involving C_4_AcH_11_, C_3_AH_6_, and AH_3_, resulting in a remarkable change on the surface of the four types of calcium carbonate. Moreover, the formed granular structure adhered to the large particles in precipitated calcium carbonate composite and <500 mesh ground calcium carbonate composites, but it cannot be clearly found in 400–500 mesh ground calcium carbonate composite. Considering that the adhered granular structure has fallen away from the surface of the calcium carbonate composites easily, the use of 400–500 mesh ground calcium carbonate is more suitable for the preparation of calcium carbonate composites.

### 3.3. Microstructure Analysis

As shown in Figure 4, with the raw calcium carbonate coated with C_4_AcH_11_, C_3_AH_6_, and AH_3_ particles, the particle size of four types of calcium carbonate composites is very large, and the traditional SEM-EDS analysis cannot detect differences in their microstructures. The microstructure of the prepared calcium carbonate composites was determined by first embedding the prepared composites in resin, polishing them to expose the internal structure, and testing them via SEM-EDS.

The SEM-EDS maps of vaterite-type precipitated calcium carbonate composite are shown in Figure 5. The element distributions of Aluminum (Al), Calcium (Ca), Oxygen (O), and Carbon (C) are obviously inhomogeneous, indicating that different species wrap each other in composite. As shown in Figure 5a, the area A has a high content of aluminum and oxygen but little content of calcium or carbon, indicating that the species phase in area A is AH_3_. Additionally, area B has a high content of calcium and oxygen and a small amount of aluminum and carbon, indicating that the specie phase in area B involves C_3_AH_6_, C_4_AcH_11_, and CaCO_3_. Furthermore, part of the perimeter of area B has a relatively high content of aluminum and oxygen, while the content of calcium or carbon is low in the perimeter of area A and in part of perimeter of area B. This finding indicates that the formation of AH_3_ prefers to adhere to the surface of C_3_AH_6_ and C_4_AcH_11_, or has independent nucleation from the solution.

Figure 6 displays the SEM-EDS maps of calcite-type precipitated calcium carbonate composite. The element distributions of Al, Ca, O, and C are also obviously inhomogeneous. As shown in Figure 6a, area A has a high content of Al and O but little content of Ca or C, while the areas B and C have a high content of Ca and a relatively high content of Al and O. Interestingly, the inner parts of areas B and C have low Al content but high Ca and O content. Therefore, the species phase in area A is only AH_3_, and those in area B and C are C_3_AH_6_, C_4_AcH_11_, and CaCO_3_, and that of CaCO_3_ is coated with C_3_AH_6_ and C_4_AcH_11_. Considering the minor differences in the elemental distributions between C_3_AH_6_ and C_4_AcH_11_, their microstructures cannot be clearly recognized. Moreover, part of the perimeter of area B and C has a relatively high content of Al and O but a low content of Ca or C, indicating that the surface of the formed C_3_AH_6_ and C_4_AcH_11_ is partly coated with AH_3_. Therefore, the microstructure of the formed calcium carbonate composite involves a three-tier structure, in which the innermost CaCO_3_ is wrapped by C_3_AH_6_ and C_4_AcH_11_, and the outermost AH_3_ adhered to the middle layer of C_3_AH_6_ and C_4_AcH_11_ with a thickness of 2–3 μm.

Figure 7 shows the SEM-EDS maps of <500 mesh ground calcium carbonate composite. It also shows that the independent precipitated AH_3_ and three-tier structure particles also appeared. The Ca-rich area has relatively low Al content, while the Al-rich area usually has relatively low Ca content. Therefore, the boundary between Ca and Al elemental distribution is relatively clear. Moreover, the inner parts of the areas do not contain Al but have high Ca and O content, indicating that inner parts only contain the specie phase of CaCO_3_. Therefore, those three-tier structure particles involve the unreacted CaCO_3_ as a core, the formed C_3_AH_6_ and C_4_AcH_11_ as an interface, and the formed AH_3_ as an overburden. However, the particle size of the CaCO_3_ core increased, and the thickness of the intermediate layer decreased from 1 μm to 2 μm.

The SEM-EDS maps of 400–500 mesh ground calcium carbonate composite are displayed in Figure 8. The elemental distribution of Ca, Al, O and C became much more regular, and the boundary between Ca and Al elemental distribution became clearer. Almost all elemental Ca is surrounded by Al, and the Ca-rich area has a very low content of Al, while the Al-rich area usually has a very low content of Ca. Additionally, both Ca- and Al-rich areas only exit in the boundary area of Ca enrichment, and the thickness of that boundary area is less than 1 μm. Based on the above XRD analysis shown in Figure 1(2), most of the particles display an obvious and complex core-shell structure with the core being unreacted CaCO_3_ and the shell involving AH_3_ and C_3_AH_6_. Notably, a few isolated AH_3_ particles were observed, as shown in Figure 8.

Based on the above SEM-EDS maps from Figure 5, Figure 6, Figure 7 and Figure 8, the microstructures of prepared calcium carbonate composites are very different from those obtained by using varied calcium carbonate as a precursor. The prepared calcium carbonate composites mainly involve three-tier structure particles and independent precipitated AH_3_ particles_._ When vaterite-type precipitated calcium carbonate was used as a precursor, many inclusions involving C_3_AH_6_, C_4_AcH_11_, and CaCO_3_ can be detected, and only a few AH_3_ particles adhere to the surface of those inclusions. When calcite-type precipitated calcium carbonate was used as a precursor, many isolated AH_3_ particles appeared and a three-tier structure with CaCO_3_ as the innermost, AH_3_ as the outermost and C_3_AH_6_ and C_4_AcH_11_ as the middle layers can be detected. Between the vaterite-type precipitated calcium carbonate composite and calcite-type precipitated calcium carbonate composite, more unreacted CaCO_3_ particles are observed, which are used as a core in the latter, and the content of isolated AH_3_ particles increased. This finding was obtained because the calcite-type precipitated calcium carbonate is more stable than the vaterite-type precipitated calcium carbonate, and the reacted vaterite-type precipitated calcium carbonate via an anti-caustic reaction may be restraining the independent precipitation of AH_3_. Furthermore, when ground calcium carbonate with a particle size of less than 500 mesh and 400–500 mesh was used as a precursor, particles with a three-tier structure became more obvious, and the particle size of unreacted CaCO_3_ core increased, while the thickness of the intermediate layer decreased with the increase in ground calcium carbonate particle size. This finding was obtained because the ground calcium carbonate is more stable than that of the precipitated calcium carbonate, and the anti-caustic reaction between calcium carbonate and sodium aluminate solution became more difficult with the increase in calcium carbonate particle size.

Considering the unavoidable anti-caustic reaction, the precipitated calcium carbonate, such as calcite or vaterite, is not suitable for the preparation of CaCO_3_/Al(OH)_3_ composites with a core-shell structure, while the ground calcium carbonate should be the best choice. The particle size of the ground calcium carbonate remarkably influenced the properties of the prepared CaCO_3_/Al(OH)_3_ composites, and the large particle size is favorable for the formation of the core-shell structure. However, considering that CaCO_3_/Al(OH)_3_ composites are used as a filler in plastic, rubber and man-made gems, the large particle size would impair other performances, such as compatibilization and reinforcement. Therefore, the use of ground calcium carbonate with a particle size of less than 400 mesh and greater than 500 mesh is suitable as a precursor for the preparation of core-shell CaCO_3_/Al(OH)_3_ composites.

### 3.4. Reaction Mechanism

Based on the above analyses, the preparation of CaCO_3_/Al(OH)_3_ composites with the heterogeneous nucleation method could be interpreted as the following process. First, when calcium carbonate is added to the sodium aluminate solution, 3CaO·Al_2_O_3_·6H_2_O (C_3_AH_6_) and Na_2_CO_3_ are easily generated [28], and this process can be represented as an anti-caustic reaction:

3CaCO_3_ + 4NaOH + 2Na[Al(OH)_4_] ⇌ 3CaO·Al_2_O_3_·6H_2_O + 3Na_2_CO_3_.(6)

Then, when more active CaCO_3_ molecules exist in the system, the formed C_3_AH_6_ may be further reacted with active CaCO_3_ to form a complex 3CaO·Al_2_O_3_·CaCO_3_·11H_2_O (C_4_AcH_11_) [29,30], which can be expressed in the following complex reaction:

CaCO_3_ + 3CaO·Al_2_O_3_·6H_2_O + 5H_2_O → 3CaO·Al_2_O_3_·CaCO_3_·11H_2_O.(7)

Finally, at the end of the above anti-caustic reaction and complex reaction, the formed solid particles become the substrate of the heterogeneous nucleation of Al(OH)_3_ [31]. The nucleated Al(OH)_3_ gradually separates out from the solution and adheres to the surface of those solid particles, thus forming CaCO_3_/Al(OH)_3_ composites. It can be represented as a seed precipitation reaction as follows:

Na[Al(OH)_4_] → Al(OH)_3_ ↓ + NaOH.(8)

The reaction mechanism of the formation of four types of calcium carbonate composites is shown in Figure 9. With the addition of raw calcium carbonate to the sodium aluminate solution, four types of calcium carbonate composites were prepared after anti-caustic reaction, complexation reaction and seed precipitation reaction. Considering the differences in the surface activity of four types of calcium carbonate, the formed CaCO_3_/Al(OH)_3_ composites displayed very different microstructure characteristics. By using vaterite-type precipitation calcium carbonate as a precursor, the prepared composite mainly contained many inclusions formed by C_3_AH_6_, C_4_AcH_11_, and CaCO_3_; using calcite-type precipitation calcium carbonate as a precursor, the prepared composite mainly involved three-tier structure particles with much smaller CaCO_3_ particles as the innermost layer, AH_3_ as the outermost layer and a very thick middle layer of C_3_AH_6_ and C_4_AcH_11_. When <500 mesh and 400–500 mesh ground calcium carbonates were used as precursors, respectively, the prepared composites mainly contained three-tier structure particles, while the particle size of the CaCO_3_ core increased and the thickness of the middle layer decreased with the increase in the particle size of ground calcium carbonate. The 400–500 mesh ground calcium carbonate is considered to be the best precursor for the preparation of CaCO_3_/Al(OH)_3_ composites with a very nice core-shell structure.

## 4. Conclusions

In the present study, four types of calcium carbonate were used as precursors, and supersaturated sodium aluminate solution was used as the substrate. CaCO_3_/Al(OH)_3_ composites with a core-shell structure were successfully prepared via heterogeneous nucleation. Considering the differences in the surface activity of the four types of calcium carbonate, the formed CaCO_3_/Al(OH)_3_ composites displayed very different microstructural characteristics. Considering the unavoidable anti-caustic reaction, the use of a precipitated calcium carbonate, such as calcite or vaterite, is not suitable for the preparation of CaCO_3_/Al(OH)_3_ composites with a core-shell structure. The particle size of the ground calcium carbonate influenced the properties of the prepared CaCO_3_/Al(OH)_3_ composites remarkably, and the large particle size is more favorable to forming a core-shell structure. Considering the application of the CaCO_3_/Al(OH)_3_ composites, the 400–500 mesh ground calcium carbonate is the best precursor for the preparation of CaCO_3_/Al(OH)_3_ composites with a very nice core-shell structure.

## Figures and Tables

**Figure 1 materials-16-00498-f001:**
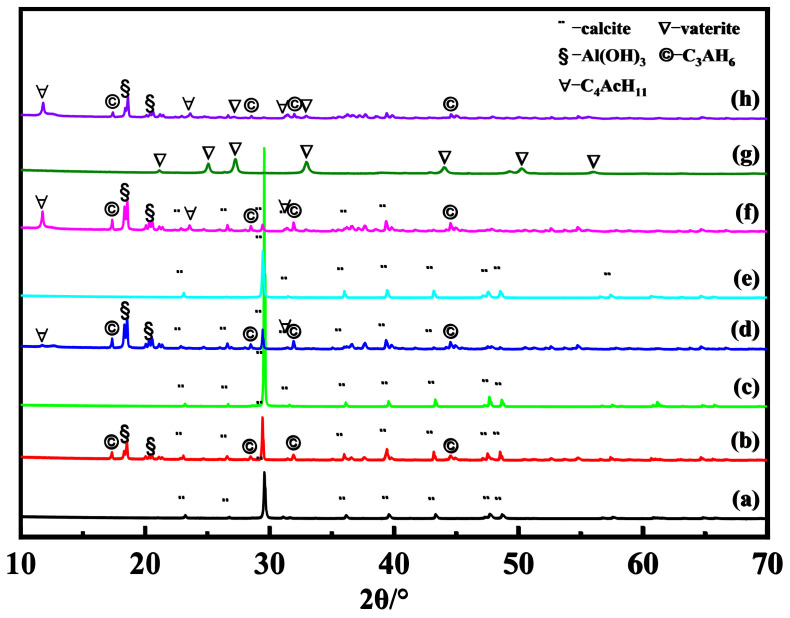
XRD patterns of different calcium sources and their composite samples ((**a**) 400–500 mesh ground calcium carbonate, (**b**) 400–500 mesh ground calcium carbonate composite sample, (**c**) –500 mesh ground calcium carbonate, (**d**) –500 mesh ground calcium carbonate composite sample, (**e**) calcium carbonate prepared from CaCl_2_, (**f**) composite sample of calcium carbonate prepared from CaCl_2_, (**g**) calcium carbonate prepared from (CH_3_COO)_2_Ca, and (**h**) composite sample of calcium carbonate prepared from (CH_3_COO)_2_Ca).

**Figure 2 materials-16-00498-f002:**
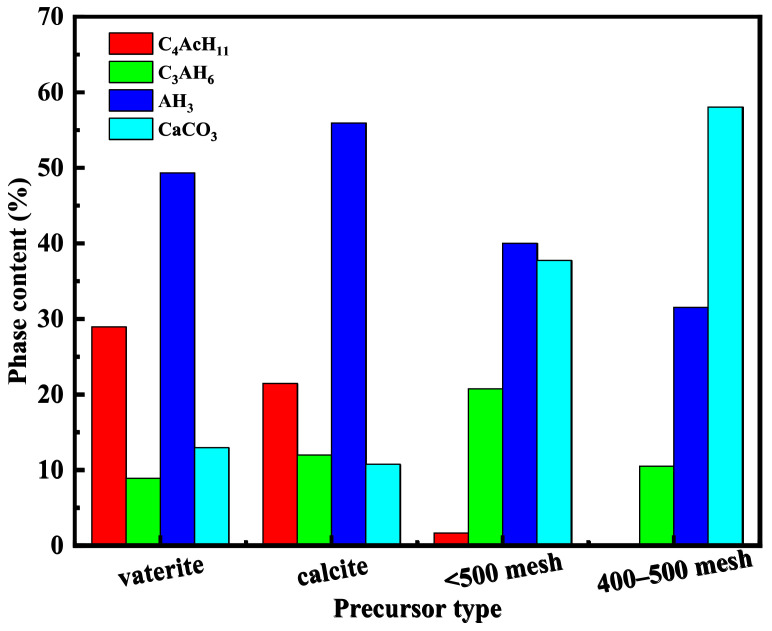
Phase content of composite samples.

**Figure 3 materials-16-00498-f003:**
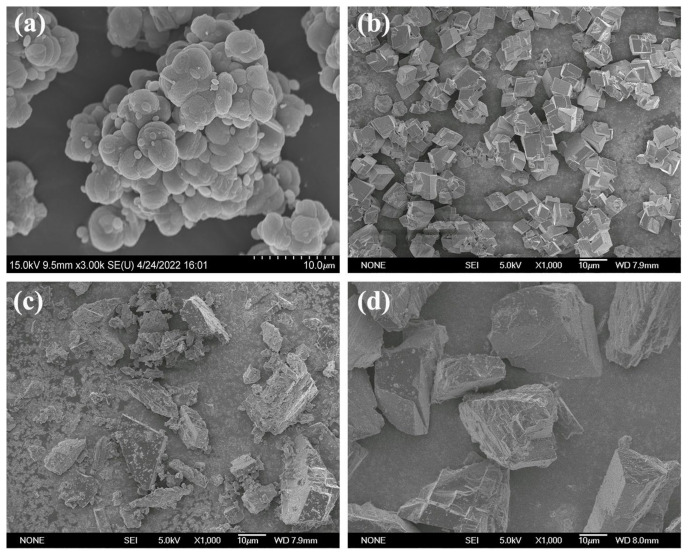
SEM images of four calcium carbonate precursors: (**a**) vaterite-type precipitated calcium carbonate; (**b**) calcite-type precipitated calcium carbonate; (**c**) <500 mesh ground calcite carbonate; and (**d**) 400–500 mesh ground calcite carbonate.

**Figure 4 materials-16-00498-f004:**
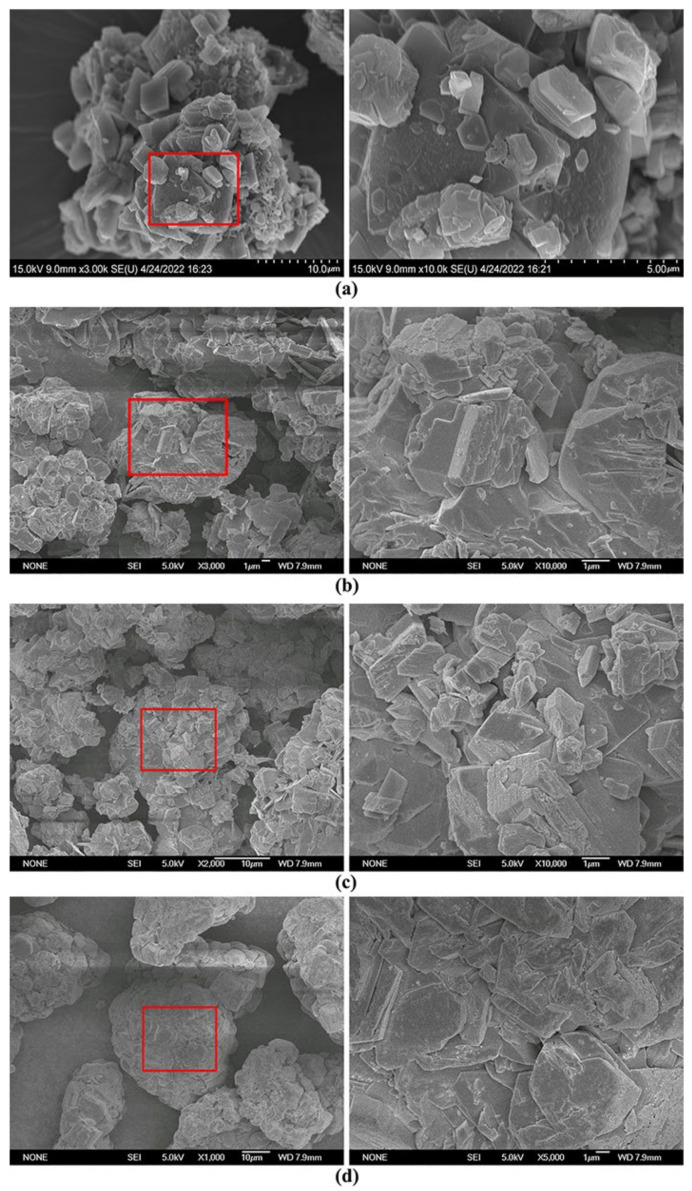
SEM images of composite samples: (**a**) vaterite-type precipitated calcium carbonate composite; (**b**) calcite-type precipitated calcium carbonate composite; (**c**) <500 mesh ground calcium carbonate composite; and (**d**) 400–500 mesh ground calcium carbonate composite.

**Figure 5 materials-16-00498-f005:**
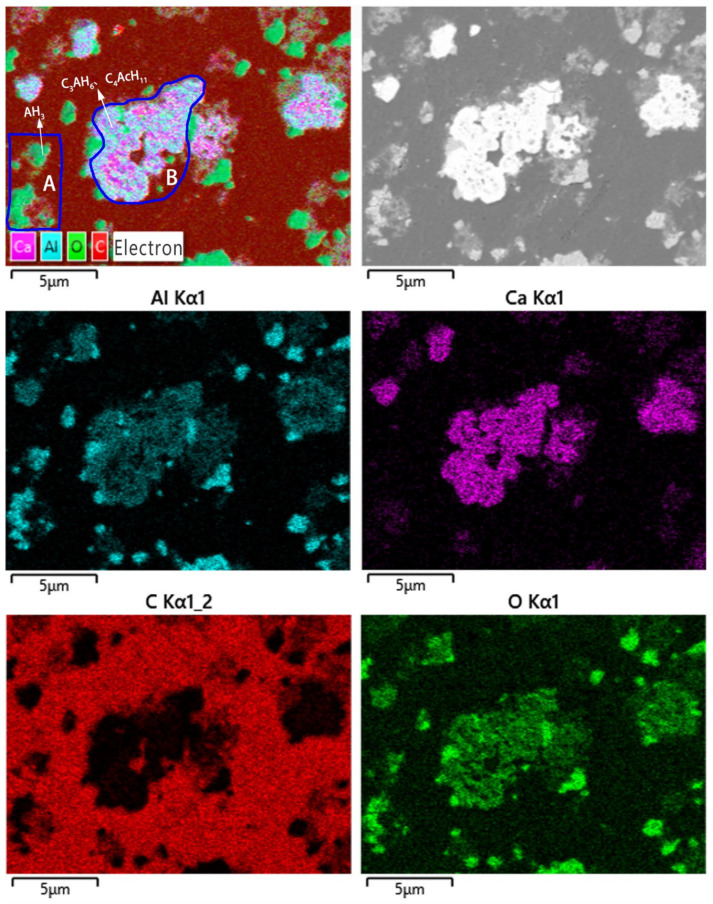
SEM image and EDS mapping of vaterite-type precipitated calcium carbonate composite.

**Figure 6 materials-16-00498-f006:**
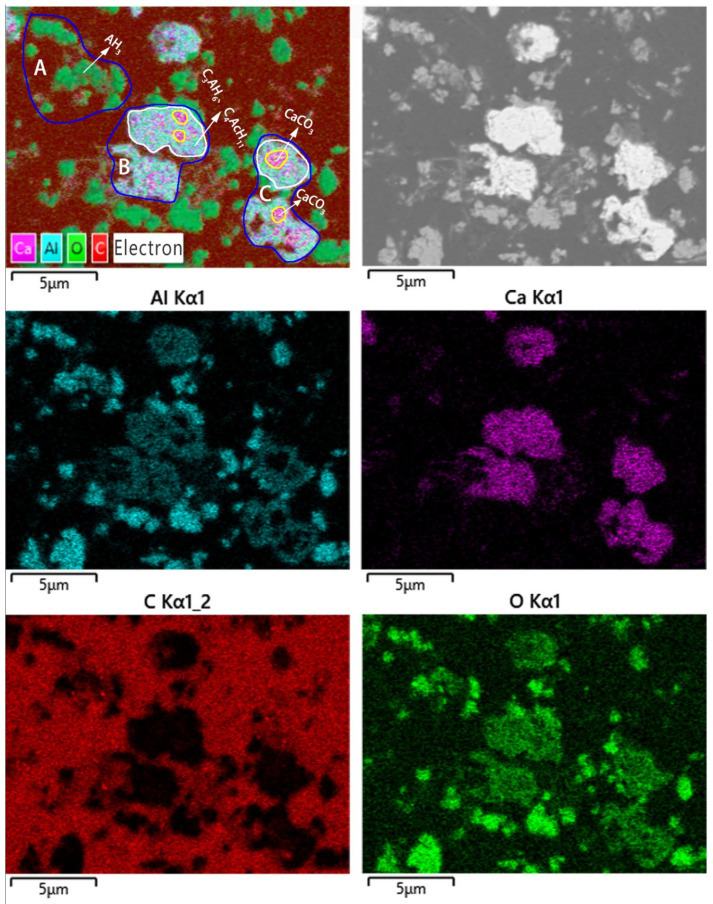
SEM image and EDS mapping of calcite-type precipitated calcium carbonate composite.

**Figure 7 materials-16-00498-f007:**
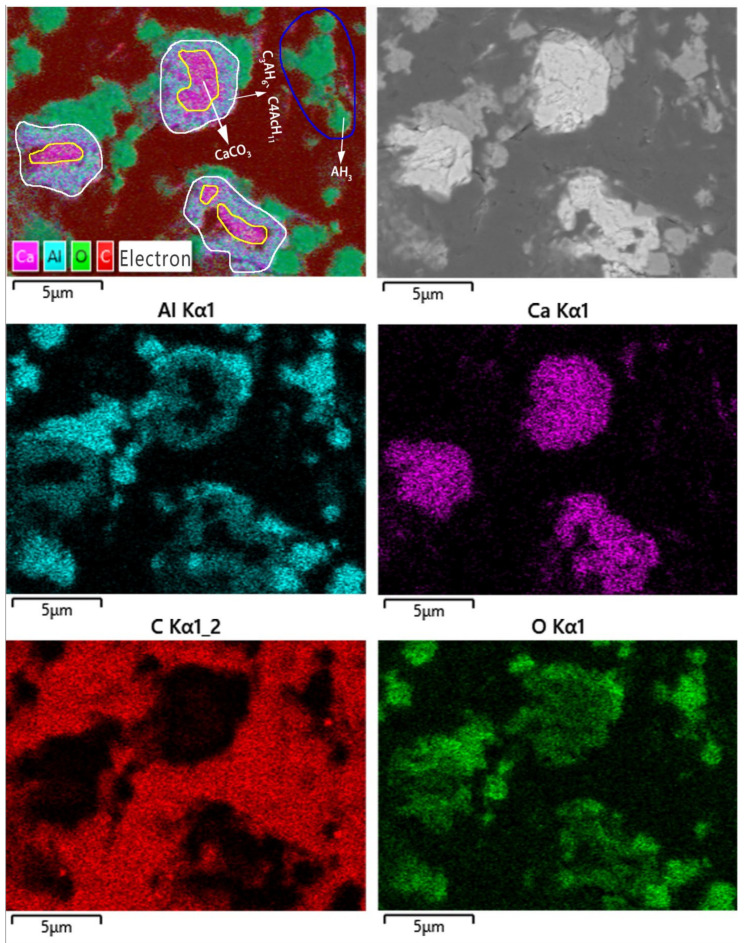
SEM image and EDS mapping of <500 mesh ground calcium carbonate composite.

**Figure 8 materials-16-00498-f008:**
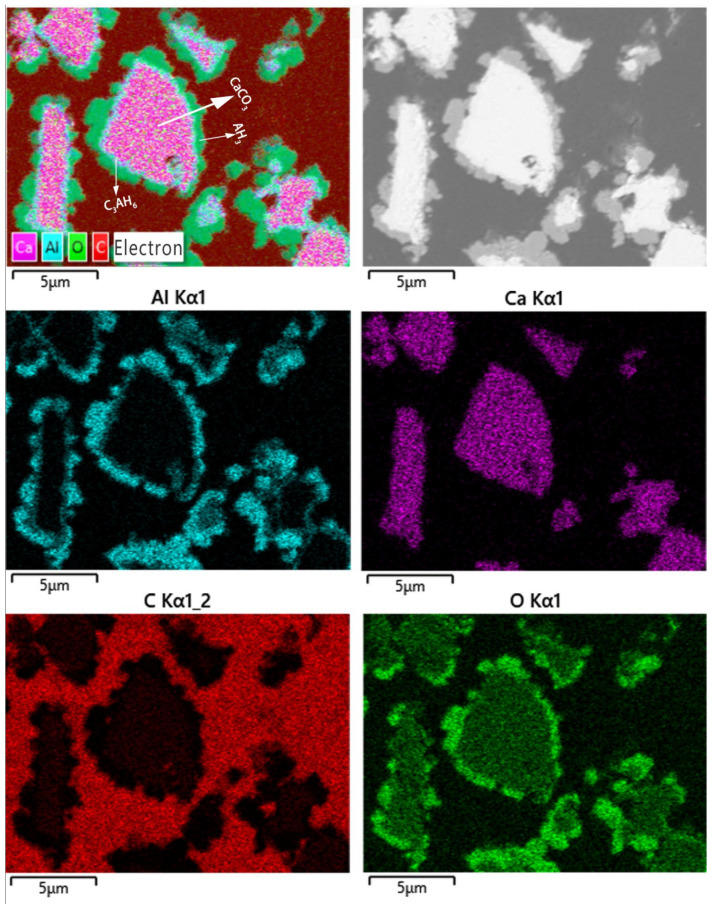
SEM image and EDS mapping of 400–500 mesh ground calcium carbonate composite.

**Figure 9 materials-16-00498-f009:**
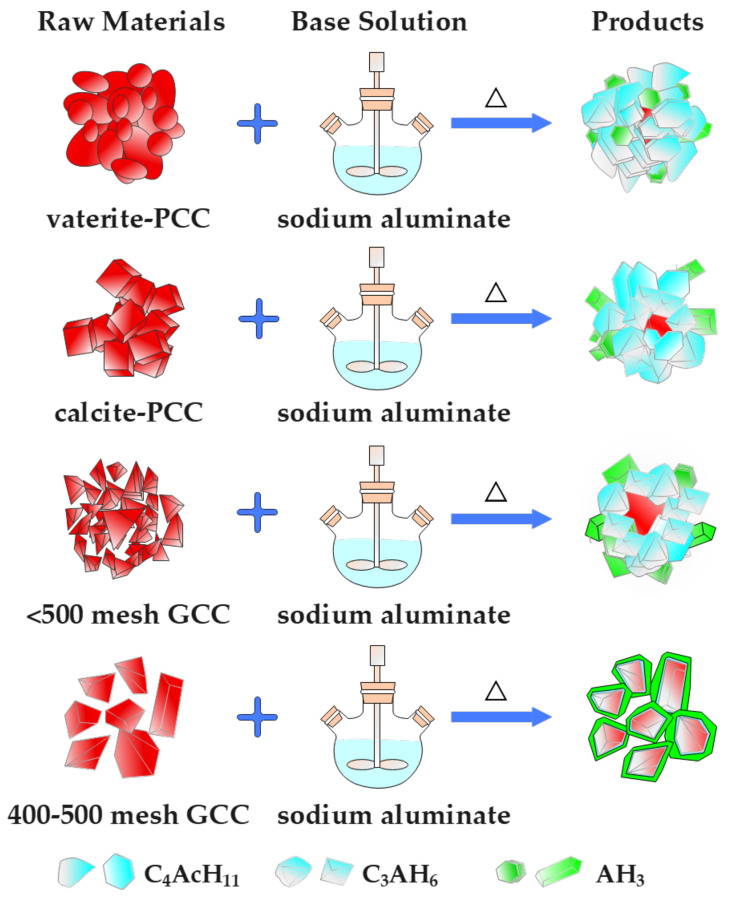
Schematic diagram of reaction mechanism.

**Table 1 materials-16-00498-t001:** Relative *K* value of phase.

Phase		File ID	Relative *K*-Value	Relative *K*-Value
C_4_AcH_11_		#87-0493	1.3557	1.7719
C_3_AH_6_		#77-0240	0.9128	1.1930
AH_3_		#85-1049	1.2148	1.5877
CaCO_3_	calcite	#99-0022	1.0	-
	vaterite	#72-0506	-	1.0

## Data Availability

The data presented in this study are available from the corresponding authors upon reasonable request.

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
