# Peer review of "Preparation of CaCO3/Al(OH)3 Composites via Heterogeneous Nucleation"

_materials, 2023, doi:10.3390/ma16020498_

Round 1

Reviewer 1 Report

Xu et al. have presented the manuscript titled: Preparation of CaCO3/Al(OH)3 Composites via Heterogeneous Nucleation. Overall presentation of the article is good, but there require few modifications before being publish, suggestions are as follow;

1.      In the abstract section, authors demonstrate that fabricated samples are best to optimize the transmittance properties, whereas the focus remained just on the characterization techniques rather on the properties, it will become a complete study if authors can add the transmittance results as well.

2.      Authors should revise the complete manuscript carefully as there exist several subscript and superscript errors in units, symbols and chemical formulas.

3.      Line 38, 39, introduction section, “The technical indexes of the white marble resources can meet the special grade level in China, in which the calcium carbonate content is above 96%, and the whiteness is above 95”, what is meant by whiteness?

4.      Line 42, English with technical language must be revised such as “The use pattern decreases the value of resource to bring out graveness waste….”. It is hard to understand what authors want to describe.

5.      In the second last paragraph of abstract, authors have described the different fabrication techniques reported by literature. Authors have strongly elaborated the values of pH for such samples. But have not described that why pH of such samples is important for this study?

6.      In the method section, please describe the manufacturer company and analytic grading of the chemicals used for this study.

7.      As authors have measured the elemental mapping for this study, there will be EDX of such samples as well. I suggest the authors to please comment on the pre and post samples fabrication about atomic wt% of constituents.

Author Response

Point 1. In the abstract section, authors demonstrate that fabricated samples are best to optimize the transmittance properties, whereas the focus remained just on the characterization techniques rather on the properties, it will become a complete study if authors can add the transmittance results as well.

Response 1:

Our objective is to obtain a more appropriate calcium carbonate raw material through characterization of the prepared CaCO3/Al(OH)3 composite materials. The composites with a core-shell structure showed a trend toward the performance of aluminum hydroxide. It can be seen which composite has obvious core-shell structure and the prepared composite has less by-products except aluminum hydroxide through the characterization of the prepared composite. According to this, a more suitable calcium carbonate raw material is obtained. With regard to the properties of CaCO3/Al(OH)3 composites, we are making active efforts to do relevant research and have obtained some results, but they are not accurate enough.

Point 2. Authors should revise the complete manuscript carefully as there exist several subscript and superscript errors in units, symbols and chemical formulas.

Response 2:

We have revised the complete manuscript carefully.

Point 3. Line 38, 39, introduction section, “The technical indexes of the white marble resources can meet the special grade level in China, in which the calcium carbonate content is above 96%, and the whiteness is above 95”, what is meant by whiteness?

Response 3:

According to the GB/T 17749-2008 methods of whiteness specification, whiteness indicates the degree of whiteness of the object color. The higher the whiteness value, the greater the whiteness. The whiteness of perfect reflecting diffuser is 100.

Point 4. Line 42, English with technical language must be revised such as “The use pattern decreases the value of resource to bring out graveness waste….”. It is hard to understand what authors want to describe.

Response 4:

We want to highlight that the use form of the high-quality marble would cause resources waste as well as unsustainable development. A new way needs to be explored for enhancing the additional value of calcium carbonate and avoiding the waste of quality resources.

Point 5. In the second last paragraph of abstract, authors have described the different fabrication techniques reported by literature. Authors have strongly elaborated the values of pH for such samples. But have not described that why pH of such samples is important for this study?

Response 5:

The reported CaCO3/Al(OH)3 composites were prepared from Ca(OH)2 as calcium source by carbonation, and the pH of the synthetic system must be precise controlled. In this paper, calcium carbonate is used as calcium source directly and no control pH value is required.

Point 6. In the method section, please describe the manufacturer company and analytic grading of the chemicals used for this study.

Response 6:

The chemicals used for this study were obtained from Sinopharm Chemical Reagent Co., Ltd., China. And they were of analytical grade. We already described in the Material and Reagents section.

Point 7. As authors have measured the elemental mapping for this study, there will be EDX of such samples as well. I suggest the authors to please comment on the pre and post samples fabrication about atomic wt% of constituents.

Response 7:

The elemental mapping for this study was obtained though area scan. The sensitivity of area scan is relatively low. It's a qualitative analysis. The atomic percentage obtained by area scan is inaccurate.

Reviewer 2 Report

Recommendation: The manuscript should be published after the authors address the Reviewer’s remarks and suggestions.

This work is devoted to the preparation and the study of CaCO3/Al(OH)3 composites. Actually after some corrections this work should be published.

My remarks:

1) Why do you not use ultrasonic treatment for preparation of these composites?

2) What do you think about co-precipitation CaCO3 and Al(OH)3 from solutions and mechanical activation of solid mixtures of CaCO3 and Al(OH)3? Have you used it? If yes, add these data, if no please explain why they are not used.

3) The manuscript needs to add information about composites testing. For example, you could test it for light transmittance. Based on the manuscript information it is not clear about consumer properties of these composites. What types of materials are better? Is there a difference between the mechanical mixture of CaCO3 and Al(OH)3 and CaCO3/Al(OH)3 composites? I assume you need to add this data to the manuscript.

Author Response

Point 1) Why do you not use ultrasonic treatment for preparation of these composites?

Response 1):

In this paper, the heterogeneous nucleation method is used to prepare composite materials, which has little relevance to ultrasonic preparation method.

Point 2) What do you think about co-precipitation CaCO3 and Al(OH)3 from solutions and mechanical activation of solid mixtures of CaCO3 and Al(OH)3? Have you used it? If yes, add these data, if no please explain why they are not used.

Response 2):

The co-precipitation preparation method requires precise adjustment of pH, and mechanical activation of solid mixtures cannot obtain the core-shell structure composite. They have little relevance to this article.

Point 3) The manuscript needs to add information about composites testing. For example, you could test it for light transmittance. Based on the manuscript information it is not clear about consumer properties of these composites. What types of materials are better? Is there a difference between the mechanical mixture of CaCO3 and Al(OH)3 and CaCO3/Al(OH)3 composites? I assume you need to add this data to the manuscript.

Response 3):

This paper reports for the first time the composite with core-shell structure prepared with calcium carbonate as the precursor and makes detailed analysis of the phase, morphology and microstructure of the composite materials. It is concluded that the composite materials prepared by ground calcium carbonate with a certain particle size as the precursor have a significant core-shell structure. In addition, composites with core-shell structure generally exhibit the physicochemical properties of the shell, which can also be proved by our preliminary experimental results. We will investigate the application properties of composite materials in further work.
